# Effectiveness of Peer-Led Wellbeing Interventions in Retirement Living: A Systematic Review

**DOI:** 10.3390/ijerph182111557

**Published:** 2021-11-03

**Authors:** Lilian Barras, Maike Neuhaus, Elizabeth V. Cyarto, Natasha Reid

**Affiliations:** 1Centre for Health Services Research, The University of Queensland, Brisbane, QLD 4102, Australia; 2Centre for Online Health, Centre for Health Services Research, The University of Queensland, Brisbane, QLD 4102, Australia; m.neuhaus@sph.uq.edu.au (M.N.); n.reid@uq.edu.au (N.R.); 3Bolton Clarke, Brisbane, QLD 4059, Australia; ecyarto@boltonclarke.com.au; 4Faculty of Health and Behavioral Sciences, The University of Queensland, Brisbane, QLD 4072, Australia

**Keywords:** peer-led, intervention, retirement living, well-being, longevity, healthy ageing, health promotion

## Abstract

Retirement living (RL) communities may be an ideal setting in which to utilize peer-leaders to implement or support health and wellbeing interventions. To date, this literature has not been systematically summarized. The purpose of this study was to fill this gap with a particular focus on describing the extent to which interventions addressed each level of the social ecological model of behavior change. This review utilized established frameworks for assessing methodological quality of studies, including the CONSORT guidelines and RoB2 bias assessment for cluster randomized controlled trials. A total of 153 records were identified from database searches, and seven studies met inclusion criteria. Overall, there is emerging evidence that peer-led health and wellbeing programs in RL communities can positively impact both health behavior, such as increased physical activity or nutrition, and health status, such as lower blood pressure. The study quality was modest to very good, but only one study was deemed not to have a high risk of bias. Peers are generally cost-effective, more accessible, and relatable leaders for health interventions that can still produce impactful changes. Future studies are needed to better understand how to sustain promising interventions.

## 1. Introduction

Older adults, aged 60 years or more, are the fastest growing age group worldwide, estimated to reach 2.1 billion people by the year 2050, or approximately 20% of the worldwide population [1]. These projections are similar in Australia, with an estimated population of 8.8 million (22%) older adults by the year 2057 [2]. Detrimental lifestyle health behavior leading to chronic diseases is recognized as a major public health concern in this age group [3]. Physical inactivity and poor dietary habits are some of the leading causes of chronic disease [4]. Physical activity reduces the risk of chronic diseases, such as hypertension, type II diabetes, and some forms of dementia [5], while inactivity has been associated with a greater risk of falls due to reduction in muscle mass and loss of balance [6]. A fall can significantly affect an older person’s independence and quality of life and could trigger admission to a high care setting, such as a residential aged care or nursing home [6]. Moreover, the mental health of older adults is a growing concern with an estimated 20% of older adults experiencing or living with mental health conditions, such as depression and anxiety [6]. Collectively, physical inactivity, poor nutrition and mental health concerns are estimated to cost the healthcare system over USD 16.6 billion annually [7,8] and are a public health intervention priority.

### 1.1. Targeting Retirement Communities

Although many interventions, particularly strength and balance exercises, are known to improve outcomes, such as function, muscle strength, and falls in older adults [9], participation rates outside of structured programs and research activities remain concerningly low. Only 25% of older adults reach at least 150 minutes of physical activity each week [10]. Although longer-term (6–12 months) maintenance is promising, effects are usually diminished and follow-up beyond 12 months is understudied [11]. Targeting particular groups during a “teachable moment” [12] may be a strategy for overcoming some of these issues. Retirement communities, for instance, are becoming an increasingly popular housing choice for older adults as the population continues to age [13]. An estimated 5.7% of older Australians currently reside in retirement communities and, over the next decade, this is projected to rise to 7.5% [14]. Retirement communities are an appealing choice as they provide independence and opportunity for social support while offering some care options [13]. They generally have communal spaces which are easily accessible to residents and also often have resources that can be utilized, including provision of recreational and social activity programs, emergency support and low maintenance living [15]. This makes them an ideal setting for wellbeing interventions. However, studies indicate residents of retirement communities are less physically active and have poorer physical function [16] and may engage in more sedentary behavior (i.e., sitting down) [17] than older adults living in the broader community. Therefore, these residents may be an ideal target population for wellbeing interventions as they are transitioning into a different living environment and adapting to a new lifestyle, potentially creating an ideal time to establish new healthy habits (3).

### 1.2. Peer-Led Interventions

Although a range of interventions have been trialled with community-dwelling older adults (from physical activity, nutrition, and social inclusion), those living in retirement villages are an understudied segment of the older adult community-dwelling population. The use of peer-leaders has become a widely used method of promoting health behaviors [18]. Peers will often share a common age, culture, knowledge of problems and bring their own community experience [18]. Social cognitive theory suggests peers are influential in promoting health behaviors because an individual is more likely to participate if the leader is a realistic figure for self-comparison [19]. The peer leader can act as a role model or a figure of hope who is able to match their support with the needs of the individual [20]. Additionally, the peer-leaders provide a low-cost alternative to trained professionals, which could allow more interventions to occur [21]. Although the success of peer-led interventions has been documented, [18,22] the research has not been systematically summarized for older adults in retirement communities. Another area of research is determining which intervention elements are linked with positive results. The social ecological model of behavior change provides an appropriate model under which to investigate the individual, interpersonal, environmental (organizational and community), and policy strategies that have been utilized [23].

The aim of this study was to systematically appraise the literature to determine the effectiveness of peer-led physical exercise, nutrition, and mental health (collectively termed “wellbeing”) interventions in retirement communities. In particular, we examined data relating to the type of interventions that were most effective, and the levels of the social ecological model they addressed. Eligible studies were examined to determine whether the interventions resulted in change both in health behavior and health indicators of older adults living in retirement communities. With a rapidly ageing population worldwide, the findings of this review will help older adults, RL providers and policy makers better utilize and support these cost-effective peer-led programs.

## 2. Materials and Methods

### 2.1. Data Sources and Search Strategy

This systematic review was conducted in accordance with the *Preferred Reporting Items for Systematic Reviews and Meta-Analyses 2020* (PRISMA-2020) guidelines [24]. The PRISMA-2020 checklist is available as Appendix A.

Literature searches of databases were conducted in August 2021 without any geographical or language restrictions using Embase, Medline, and PubMed. Only peer-reviewed and published literature was included. Grey literature was not included in our search. The search terms and strategy used in this review are described in Table 1 and resulted in 141 peer-reviewed publications. This review was registered with PROSPERO (*CRD42021240589)* [25]. A review protocol has not been previously published.

Supplementary searches of the reference lists, forward citations of the eligible papers, and the authors’ own literature databases were also conducted. Titles and abstracts were screened and full text reviews were conducted by two reviewers independently (LB and NR). Disagreements were resolved through discussion.

### 2.2. Type of Intervention

This review included papers studying peer-led wellbeing interventions in retirement communities. This approach allowed a range of interventions focusing on physical and mental wellbeing to be included, such as walking groups, physical activity, and nutrition programs, osteoporosis education and exercise, exercise groups focusing on cognitive outcomes and function focused care. In some studies, the interventions included group education and support sessions and utilized a range of other educational tools, such as pamphlets and charts. The primary outcome for each intervention was to improve the wellbeing of residents in the retirement community. All studies also had a significant component of the intervention facilitated by a peer leader.

### 2.3. Inclusion/Exclusion Criteria

To be included in this review, papers had to be publications of original research (i.e., not a review or protocol paper), written in English, and describe the outcomes of an intervention. The intervention or components of the intervention had to be peer-led and the retirement community setting was necessary. This included assisted living facilities which provide a lower level of care than nursing homes. A peer was defined as a person of a similar age and sociodemographic profile as the intervention participants who had a major role in teaching or facilitating the intervention. Peers must have undergone some form of training. The search was open to any wellbeing intervention in the retirement community setting. Studies were not restricted by participant number or length of study. There was no exclusion due to age or comorbidities of participants as the individual papers had their own restrictions depending on the intervention. Due to the expected low number of eligible published papers, papers were eligible if they were based on randomized controlled trials (RCTs) or cluster-RCTs, controlled and uncontrolled before–after designs.

Studies were only excluded if they were not peer-led, not based in a retirement community, and not an original study reporting an intervention. Studies were also excluded if they did not report on health behavior or health status at all.

### 2.4. Study Selection

The study selection PRISMA flow diagram is shown in Figure 1 Titles were imported into the *Covidence Systematic Review Software* (Veritas Health Innovation, Melbourne, Australia. Available at www.covidence.org, accessed on 15 January 2021) to conduct all screening and data extraction. The search identified 141 publications, with an additional 12 studies added from reference list searchers (*n* = 11), and forward citations (*n* = 1). A total of 124 studies were excluded, either as duplicates (*n* = 10) or not meeting the inclusion criteria at the title/abstract screening stage (*n* = 114). A total of 29 studies underwent full text review, seven of which met all inclusion criteria.

### 2.5. Reporting Quality and Risk of Bias Assessment

In order to assess reporting quality, we used the Consolidated Standards Of Reporting Trials (CONSORT) statement, a validated tool comprising a checklist of 25 items [26]. We confirmed the adherence to the 25 items and scored each item as: 0 (no adherence) or 1 (full adherence). The final CONSORT-based score achieved by each study was determined as a percentage of the maximum possible score. Questions related to randomization, for example, were excluded when considering non-randomized trials. After the exclusion of any potential items scored as ‘Not Applicable’, we presented the CONSORT-based score as a percentage of the total possible.

To assess risk of bias, we used the Cochrane Handbook for Systematic Reviews of Interventions [27], particularly the Risk of Bias 2 (RoB2) assessment tool extension for cluster randomized trials. This tool allows review authors to make judgements about the risk of different sources of bias (e.g., selection, performance, detection, attrition, reporting and other bias), and succinctly summarize their evidence for this bias in comments for each study. Two authors (NR and MN) independently assessed each study using the RoB2 tool and assigned a high, low, or some risk of material bias for each item for each study included in the review, where material bias refers to bias of a magnitude to have a notable effect on the results or conclusions of the trial [27].

### 2.6. Data Extraction and Synthesis

For each of the seven included studies, we extracted the following data: details about country in which the study was conducted, number of participants, aim of the study, description of the included sample (e.g., % female, age, characteristics), description of the intervention and outcomes measured, and intervention results. We also extracted an overview of the studies and interventions, including how levels of the social ecological model of behavior change were addressed. Finally, we summarized the effectiveness of each study in (1) changing intended behavior (e.g., increasing physical activity, improving diet quality), (2) changing health indicators (e.g., increasing muscle strength, improved blood pressure, etc.). Authors of included studies were contacted to ascertain effect sizes if corresponding data were not available in the original text. Effect measures were mostly presented as mean differences from pre- to post-intervention. No data conversions were necessary to describe the results.

## 3. Results

### 3.1. Overview of Studies

The seven manuscripts included in this review describe results from six distinct programs. A pre-post intervention trial design was used in two of the included studies. One was from a program called *Project Healthy Bones* (PHB) [28], which aimed to investigate the applicability of PHB previously delivered in community settings to the assisted living population. The other was an implementation study of the *Function Focused Care—Assisted Living* (FFC-AL) [29] trial. The remaining four studies were cluster-randomized controlled trials, including the *Multilevel Intervention for Physical Activity in Retirement Communities* (MIPARC) [3,30], the *Residents in Action Trial* (RiAT) [31], the *Retirement Village Physical Activity and Nutrition for Seniors* (RVPANS) [14] program, and the original FFC-AL cluster-RCT [32]. In the RiAT study, recruitment of peer-leaders (termed ‘ambassadors’) was insufficient, turning the planned RCT into a quasi-experimental design. The total number of unique participants from all studies in this review was 4809, ranging from 40 [28] to 3676 participants [29]. Females represented 70% or more of the population in each study and mean age ranged from 72 to 83 years. To be eligible for most of the studies, participants had to be under-exercising according to guidelines while also not exhibiting any contraindications to physical activity, such as being at risk of falls or cardiovascular complications. In some studies, potential participants had to pass tests to meet the criteria, such as a mobility test. Therefore, the target population for these trials were generally healthy residents of retirement communities who were underactive.

### 3.2. Reporting Quality and Risk of Bias Assessments

#### 3.2.1. Reporting Quality

The adherence of studies in this review to CONSORT items ranged from 48% to 92%. Appendix A presents the reporting quality of the included studies for each item of the CONSORT checklist.

#### 3.2.2. Risk of Bias in Cluster RCTs

The RoB2 tool for cluster RCTs was used to assess risk of bias for the four cluster-RCTs included in this review. Overall, all but the FFC-AL study were classified as high risk, based on having at least one type of bias in this category (Figure 2). Risk of bias due to the timing of identification and recruitment of participants was identified as high risk in two studies [3,14]), due to the fact that recruitment occurred after randomization of clusters. The RiAT study was found to have the highest risk based on bias due to deviations from the intended intervention (in terms of assignment and adherence) and missing outcome data.Appendix A displays the individual risk of bias assessment for each study.

##### Qualitative Assessment of Bias in Pre-Post Studies

No specific tool was used for bias risk in the PHB and FFC implementation studies, although a narrative appraisal is summarized here. The PBH trial excluded participants with moderate to severe cognitive impairment (based on cognitive screening), and those without medical clearance. The analyses utilized also did not account for any confounding variables. However, as a pilot study with data on only 40 participants, these practices are within norms. Many outcome variables were based on performance tests, with minimal issues arising from self-reported data. The FFC implementation study predominantly focused on dissemination outcomes, such as reach, adoption, and implementation.

### 3.3. Description of Interventions and Target Outcomes

Table 2 below provides a summary of the studies and any findings of note. Supplentary Appendix A provides a detailed overview of each study, including a description of their interventions, target populations, outcome measures, and findings. All interventions in this review utilized peer-leaders (alternatively referred to as champions or peer-mentors or ambassadors) to deliver the intervention materials. All involved some form of educational materials, group workshops, or peer motivational strategies, while some trials additionally focused on community, environmental, or policy level strategies [3,29,32]. The education component was slightly different in each trial, but generally aimed to teach participants about goal setting and the benefits of meeting recommended guidelines for physical activity. The primary behavior change outcome in the majority of trials was increased physical activity. Three of the six trials measured physical activity with an objective monitor [3,30,31,32], while one trial used a validated self-reported questionnaire [14], and the two pre-post studies did not measure physical activity [28,29]. The RVPANS study also aimed to influence nutrition and dietary behavior [14]. At least one health indicator was included in all trials such as, but not limited to, measures of physical function (i.e., muscle strength, timed up and go test), falls, weight/body mass index, blood pressure, and cognitive function.

#### Intervention Parameters

All studies included intervention paramaters addressing the intra- and inter-personal levels of the social ecological model. Individual goal-setting and motiviational interviewing are strategies that target the intrapersonal level of the social ecological model. The inclusion of peer-leaders as those delivering the programs in all studies addresses the interpersonal level of the model. Beyond this, the MIPARC and FFC trials were the only programs to address more levels of the social ecological model, including environment, community and policy. In the MIPARC study, peer-leaders advocated for environmental improvements in order to facilitate safe walking conditions and other means of active commuting. Changes included extending cross-walk times, including both visual and auditory traffic cues, and cutting back foliage across sidewalks. Leaders received training from a local community advocacy organization on how to communicate with local policy makers. Environmental changes took six months to implement on average. From a community perspective, peer-leaders brought together other participants to advocate for these changes [3,30]. The FFC-AL intervention employed similar techniques across the ecological framework, with a major difference being that peer-leaders were not residents themselves, but rather, key staff members, specially trained nurses and carers [32]. Intervention duration ranged from 16 weeks [31] to 12 months [3,32].

### 3.4. Effectiveness of Interventions to Change Health Behavior

All four cluster-RCTs measured change in physical activity following their interventions, and all four found positive results. In both the MIPARC [3,30] and RVPANS [14] trials, the intervention group increased their physical activity compared to the control groups. The FFC-AL RCR saw positive trends in this direction, although they did not reach statistical significance [32]. Participants in the main intervention group in the RiAT trial [31] improved their physical activity (measured as overall stepping time) compared to baseline, although between group changes were not reported. Regarding other positive health behaviors, the RVPANS trial found improvements in the intervention group for increased fruit and fibre intake, less sitting time and more participation in strength training [14].

### 3.5. Effectiveness of Interventions to Change Health Indicators

The RVPANS study reported a statistically significant improvement in at least one health indicator, specifically a reduction in weight and diastolic blood pressure [14]. In the FFC-AL RCT, physical function (measured with the Barthel Index) declined in both the intervention and control groups. However, by 12 months the decline was greater in the control group compared with the intervention group (*p* = 0.01) [32]. Differences in other outcome variables, such as depression, resilience, and BMI, were not observed in these trials. Similarly, there were no statistically significant differences in health indicators in the RiAT and MIPARC RCT main outcomes [3,31] (see Supplemental Appendix A for all details).

In the non-randomized trials, participants in the PHB trial significantly improved their strength, balance, posture, and flexibility compared to baseline [28]. Likewise, in the implementation study of the FFC-AL trial, a statistically significant reduction in number of falls was reported after the intervention [29].

Zlater et al. (2019) [30] assessed the effect of physical activity on cognitive ability within the MIPARC study. Although no intervention effects on participants’ cognition were observed, those who increased moderate-intensity physical activity by 10 mins/day (whether in the control or intervention group), showed a statistically signficiant improvement of 5%–7% in tests of cognitive flexibility and executive function, and a 0.82-point improvement in symbol search scores, a measure of psychomotor speed and visual scanning.

## 4. Discussion

The aim of this study was to systematically review the literature on the effectiveness of peer-led interventions to improve the physical and mental wellbeing of older adults living in retirement communities. A total of seven publications describing outcomes from six separate trials were included, covering data from 4809 unique participants. Overall, an improvement in health behavior was observed in all four RCTs. Improvement in at least one health status indicator was also observed in three of the RCTs, and in both pre-post studies. Reporting quality in these studies ranged from average to very good, while three of four RCTs were judged to be at high risk of bias. Our review indicates that, overall, there is some evidence showing that peer-led interventions in retirement communities can improve residents’ health and wellbeing; however, more studies with better quality of evidence are required to make further judgements.

The results from this systematic review are broadly consistent with literature regarding other peer-led interventions for older adults. Grysztar and colleagues determined in their systematic review that five of seven intervention studies were successful in promoting health practices to older adults [22]. In Burton’s 2017 systematic review evaluating the effectiveness of peers at motivating older people to increase physical activity, 16 of the 18 studies included found an increase in physical activity [33]. This review also determined that peers helped to promote and maintain adherence to the programs in community-dwelling adults aged 60 years or more. However, the Burton review also reported that meta-analyses of intervention effects on the physical function tests showed that control group participants actually performed better [33]. These findings are contrary to expectation, but based on only a handful of studies and need further assessment. Peer-led programs need to ensure that leaders are supported through their leadership process and receive appropriate training [34].

Arguably, the most successful interventions assessed in this review were the RVPANS and MIPARC trials, which reported increased physical activity [3,14], participation in strength training, and fruit intake [14], and improvements in health indicators including weight and blood pressure. Both of these studies were six months in duration (with a further six months follow-up in the MIPARC study), provided tailored support based on goal setting and motivational interviewing for the residents, and were multi-dimensional or choice-based, allowing residents to have autonomy over the exercises they chose to participate in. Although the RiAT study also considered levels of the social ecologic model, and engaged residents in goal setting and self-monitoring, it differed to the other programs in that it was prescriptive (residents were required to attend walking groups at certain times led by peers), and shorter (16 weeks). Lastly, the FFC-AL trial varied significantly from the other programs in that it was aimed at staff and carers, with a particular focus on environmental and policy changes. In this program, positive effects on residents’ health behaviors and status were evident at 12 months.

Challenges with these programs have been identified. Firstly, recruitment of peer leaders and participants was difficult. Some authors pointed out that active recruitment strategies (such as phone calls and speaking with specific people, rather than posters, for example) could be more successful in reaching a higher proportion of residents [35]. Secondly, as is the case in the majority of health interventions, the bulk of participants were females. Identifying appropriate messaging and targeting strategies to engage male participants and peer-leaders is a priority of future research in this area. Further, when asked, some peer-leaders noted concerns about the amount of time they would need to invest in their roles [31]. Developing materials, resources, and training modules to assist peer-leaders in their roles may be an option in mitigating the perceived time requirements. Lastly, buy-in from those in leadership roles within organizations has been identified as a key feature required for succesful implementation of these programs.

The findings of this review need to be viewed in light of the limitations of included studies. The change in health behavior could potentially be due to reactivity or selection bias. Participants may have changed their behavior due to their knowledge of the intervention and measurement devices and, as a consequence, positively skewed the results. Although improvements in health behaviors were observed during the study period, conclusions regarding long-term behavior change cannot be made. Additionally, selection bias could have impacted the results. Those who are willing to participate in physical exercise interventions may already be aware of the benefits of exercise and have been exercising throughout their life. Those most likely to participate in the intervention are potentially already more inclined to change their behavior. This would result in the findings inaccurately representing the true relationship between the intervention and outcome within the average retirement community population. Most participants were female, white, and had completed tertiary education. Lastly, our search did not include grey literature reports, which may detail results from other studies implemented by RL providers themselves that were not published in peer-reviewed journals. Future studies should include participants with a wider range of demographic characteristics to increase the generalizability of the interventions. Importantly, we need to understand the effectiveness and safety of these interventions in more frail individuals, who are underrepresented in similar studies.

## 5. Conclusions

Peers are generally cost-effective, more accessible and relatable leaders for health interventions, which often provide as much benefit as health professional led interventions in certain circumstances. Appropriate training of peer leaders is important, and use of motivational interviewing and choices of intervention activities are important. Based on the characteristics of the most successful programs reviewed in this study, any stakeholders interested in promoting peer-led wellbeing programs in RL communities should favor programs of six months duration or more, targeting multiple levers (from consumers to management, policy, and culture), and a focus on incorporating activities and healthy behaviors that are important and meaningful to participants. Programs that are prescriptive or offer only one type of activity are unlikely to be successful for a large proportion of participants. Future research is needed to better understand how to successfully recruit both peer-leaders and participants, how to engage retirement village managers and ensure sustainable implementation of these programs beyond the research studies.

## Figures and Tables

**Figure 1 ijerph-18-11557-f001:**
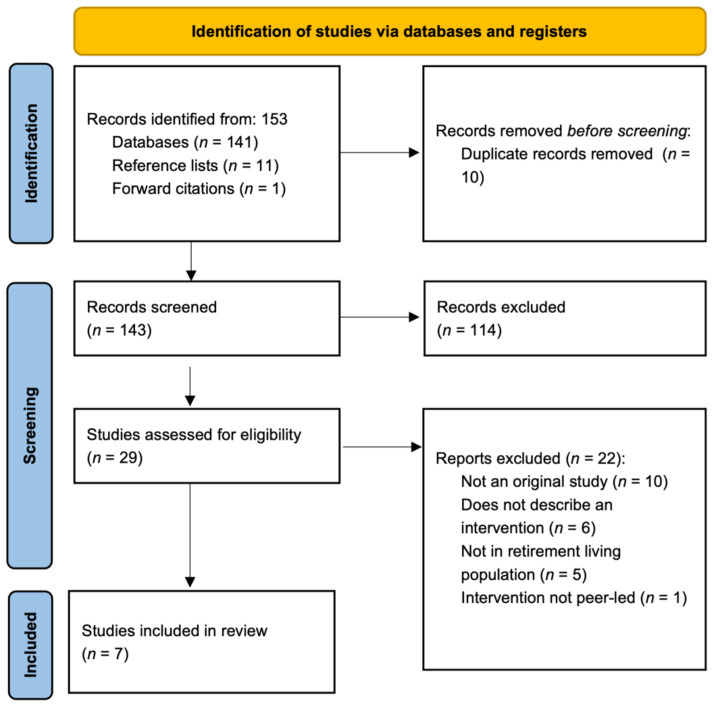
PRISMA flow diagram.

**Figure 2 ijerph-18-11557-f002:**
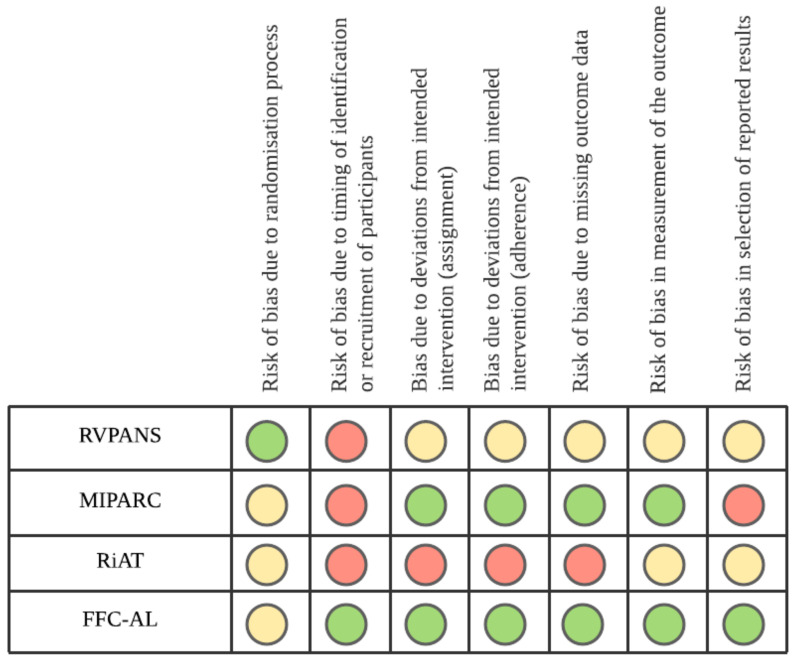
RoB2 (Risk of Bias 2 ) assessment of bias for cluster RCTs (randomized controlled trials). Retirement Village Physical Activity and Nutrition for Seniors (RVPANS); Multilevel Intervention for Physical Activity in Retirement Communities (MIPARC); Residents in Action Trial (RiAT); Function Focused Care—Assisted Living (FFC-AL). Green = Low Risk of bias; Yellow = Unclear Risk of bias; Red = High Risk of bias.

**Table 1 ijerph-18-11557-t001:** Summary of search terms and search strategy for the databases Embase, Medline, and Pubmed.

“peer support” or “peer mentoring” or “peer based” or “peer led” or “peer education” or “peer run” or “peer taught” or “peer group” or “peer counselling” or “peer-led” or “peer-based”
2. “retirement” or “retirement home” or “retirement village” or “retirement communit*” or “retirement cent*” or “assisted living” or “supported living”
3. “intervention” or “wellbeing intervention” or “well-being intervention” or “well being intervention” or “exercise intervention” or “physical activity intervention” or “exercise program” or “nutrition intervention” or “diet intervention” or “health intervention” or “lifestyle intervention or “behavio* change intervention” or “wellness activit” or “strength training”.
4. (1) AND (2) AND (3)

**Table 2 ijerph-18-11557-t002:** Summary of results of included studies.

Program	*n*	Design	Main Results
PBH	40	Pilot pre-post	Better performance on: Timed up and go Test (mobility and dynamic balance) Four square step test (balance) Tandem stand (balance) Weight lifted—arms (strength) Weight lifted—legs (strength)
FFC-AL	3676	Pre-post	Reduction in falls
FF-AL	171	Pilot RCT	Improvement in physical function (Barthel Index) at 12-months
RiAT	116	Quasi-RCT	Higher daily step count and total stepping time
RVPANS	363	Cluster RCT	Decreased: Weight and diastolic blood pressure Sitting time Increased:Engagement in strength training Fruit intake
MIPARC	307	Cluster RCT	Increased participation in moderate-to-vigorous physical activity, and improved blood pressure. Increases in physical activity, regardless of group, led to improvements in cognition

## Data Availability

Not applicable.

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
