# Peer review of "Effectiveness of Peer-Led Wellbeing Interventions in Retirement Living: A Systematic Review"

_ijerph, 2021, doi:10.3390/ijerph182111557_

Round 1

Reviewer 1 Report

Dear colleagues, I hope this message find you well.

Thank you for giving me the opportunity of reading the work “Effectiveness of Peer-Led Wellbeing Interventions in Retirement Living: A Systematic Review, it has been a very big pleasure to collaborate reviewing this manuscript. The topic of this paper is very interesting and it seems necessary to delve it. I strongly recommend the publication, however, there are several some minor issues to address before to do it:

  • It is advisable to do a final proofreading to avoid typographical mistakes as: ecologicl (line 17).

Introduction:

  • I recommended to divide the introduction into several subsections in order to facilitate the text comprehension.

Method

  • Figure 1 should be written using the same font (Palatino)

Discussion

  • It is necessary to describe in more detail the practical implications. In your opinion, ¿which could be the most adequate activity to motivate elder people to interact with their peers? ¿ How could this work help to design social policies for retired people?

Author Response

Question

Response

Reviewer 1

I recommended to divide the introduction into several subsections in order to facilitate the text comprehension.

We have added two subsections to the introduction, at lines 47 and 70

Figure 1 should be written using the same font (Palatino)

The font has been changed to Palatino

It is necessary to describe in more detail the practical implications. In your opinion, which could be the most adequate activity to motivate elder people to interact with their peers? How could this work help to design social policies for retired people?

We have updated the conclusion of this manuscript, starting on line 395, to better reflect the practical implications of the findings in this review. Our focus was on interventions that motivate people to participate in health enhancing activities, with the use of peers as facilitators.

Many other practical implications have also been detailed in a paragraph from lines 354 to 365.

Reviewer 2 Report

Dear Authors,

I want to thank you for giving me the opportunity to review the enclosed manuscript for the International Journal of Environmental Research and Public Health (IJERPH). From my inspection of the enclosed manuscript entitled “Effectiveness of Peer-Led Wellbeing Interventions in Retirement Living: A Systematic Review”, I am recommending a “Minor Revisions” decision by IJEPR due to methodological and some expository issues.

In short, my primary concern pertains to the presentation of Results. In-text, I questioned how many studies may have been excluded for “non-primary” outcomes. Given the relatively small number of studies included (k = 7), it seems that, if a large number of studies were excluded simply because the SysRev´s focus was not the primary outcome, then this may be overly narrow/restrictive. If more than 3 studies fit this criteria, I recommend authors report this in tabled format with a brief description of findings as supplementary appendix.

Generally, this  SysRev is uniquely contributive, and few could doubt the authors comprehensive application of best-practice standards for the methodology and bias appraisal. Retirement researchers may find this systematic review insightful and informative.

Again, it has been a pleasure to consider your submitted manuscript to the IJERPH – I would gladly review a resubmission.

With kind regards and cordially yours,

Reviewer

Author Response

Reviewer 2

In short, my primary concern pertains to the presentation of Results. In-text, I questioned how many studies may have been excluded for “non-primary” outcomes. Given the relatively small number of studies included (= 7), it seems that, if a large number of studies were excluded simply because the SysRev´s focus was not the primary outcome, then this may be overly narrow/restrictive. If more than 3 studies fit this criteria, I recommend authors report this in tabled format with a brief description of findings as supplementary appendix.

We thank the reviewer for this very important point. We understand that our explanation of inclusion criteria wasn’t clear. Studies were not excluded if their primary outcome didn’t match our criteria. We included all peer-led wellbeing interventions in retirement villages. We have clarified this point as follows:

Line 145:  Studies were only excluded if they were not peer-led, not based in a retirement community, and not an original study reporting an intervention. Studies were also excluded if they did not report on health behaviour or health status at all.

Further, we also recognize that there may be other studies in the grey literature that we did not include in this review. This has been acknowledged within the manuscript as follows:

Methods

Line 111: Grey literature was not included in our search.

Discussion

Line 390:  Lastly, our search did not include grey literature reports, which may detail results from other studies implemented by RL providers themselves that were not published in peer-reviewed journals.

Reviewer 3 Report

  1. This manuscript aims to summarize the literature regarding effectiveness of peer-led wellbeing interventions in retirement living. Even though I recognize the importance of systematically summarize this literature, I have some comments and questions that I describe below.
  2. My greatest concern is with the contribution of the present research. Moreover, I consider that the interest of the subject is not introduced well enough.
  3. Please provide additional information regarding the inclusion criteria of the studies that were included in this review.
  4. In my opinion, the theoretical rationale that you offer is incomplete and could/should be more deepen.
  5. Please develop the theoretical implications of the research.
  6. The implications for practice should also be more developed and clearly stated.
  7. I suggest you take another pass through the manuscript to clean up minor grammar and usage issues. Otherwise, the manuscript reads well.

Author Response

Reviewer 3

My greatest concern is with the contribution of the present research. Moreover, I consider that the interest of the subject is not introduced well enough.

We appreciate the reviewers concerns with regards to the interest of this subject to all readers. However, given it has been submitted to a special issue specifically targeting peer-led wellbeing interventions, we hope the editors will see it’s applicability.

To emphasize the importance of peer-led programs in the future and why this review is important, we have added a line at the end of the introduction, stating:

Line 100:  With a rapidly ageing population worldwide, the findings of this review will help older adults, RL providers and policy makers better utilize and support these cost-effective peer-led programs.

Please provide additional information regarding the inclusion criteria of the studies that were included in this review

We have provided detailed inclusion/exclusion criteria in the manuscript, starting at line 131. Please also refer to our response to reviewer 2 above, providing more details regarding these criteria.

In my opinion, the theoretical rationale that you offer is incomplete and could/should be more deepen.  Please develop the theoretical implications of the research

We would be very happy to provide more details to satisfy the reviewer; however, we are unclear about what theoretical implications they refer to and what further details are requested.

The implications for practice should also be more developed and clearly stated.

Thank you for your comment. Please refer to our response to reviewer 1 above, regarding practical implications of this review.

I suggest you take another pass through the manuscript to clean up minor grammar and usage issues

We have proof-read the entire manuscript and have made minor adjustments as necessary. Please see tracked changes throughout for details.
